

# A time-dependent Hartree-Fock study of triple-alpha dynamics

**Paul D. Stevenson⋆ and J. L. Willerton**

Department of Physics, University of Surrey, Guildford, GU2 7XH, UK

⋆ p.stevenson@surrey.ac.uk

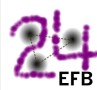

## Abstract

**Time-dependent Hartree-Fock calculations have been performed for fusion reactions of $^4$He + $^4$He → $^8$Be*, followed by $^4$He + $^8$Be*. Depending on the orientation of the initial state, a linear chain vibrational state or a triangular vibration is found in $^{12}$C, with transitions between these states observed. The vibrations of the linear chain state and the triangular state occur at ≃9 and 4 MeV respectively.**

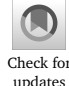
## 1 Introduction

The triple-alpha reaction is established as the process by which helium is first burned to form heavier elements in stars [1]. It is a two-stage process in which two $^4$He nuclei combine to form a short-lived $^8$Be nucleus, which later reacts with a further $^4$He nucleus to form $^{12}$C. In the stellar environment most $^8$Be nuclei created through alpha fusion disintegrate back to two alpha particles, but a small equilibrium concentration of $^8$Be allows some triple-alpha reactions to proceed. Understanding the details of the reaction is crucial to understanding this key bottleneck reaction in stars, and is a test of nuclear models which need to produce the right structures in $^{12}$C in order to describe the reaction well [2, 3].

We use time-dependent Hartree-Fock to analyse this reaction. Our work is similar to a previous TDHF study of the triple-alpha collision [4, 5], though our analysis and methods differ somewhat in its use of a two-step process and the spectral analysis of the compound nuclei. We note, too, work in which the excitation of shape isomers in the $6\alpha$ $^{24}$Mg nucleus were studied [6], which has bearing on the use of TDHF for alpha cluster states, and on other recent work on He nuclei in TDHF calculations [7], which serve to validate the general method.

## 2 Methodology

Our calculations use time-dependent Hartree-Fock (TDHF) [8] with the Skyrme interaction [9] with an unmodified version of the Sky3D code [10, 11]. We use the SLy4d interaction [12] which was fitted with no centre of mass correction as ideal for TDHF calculations.

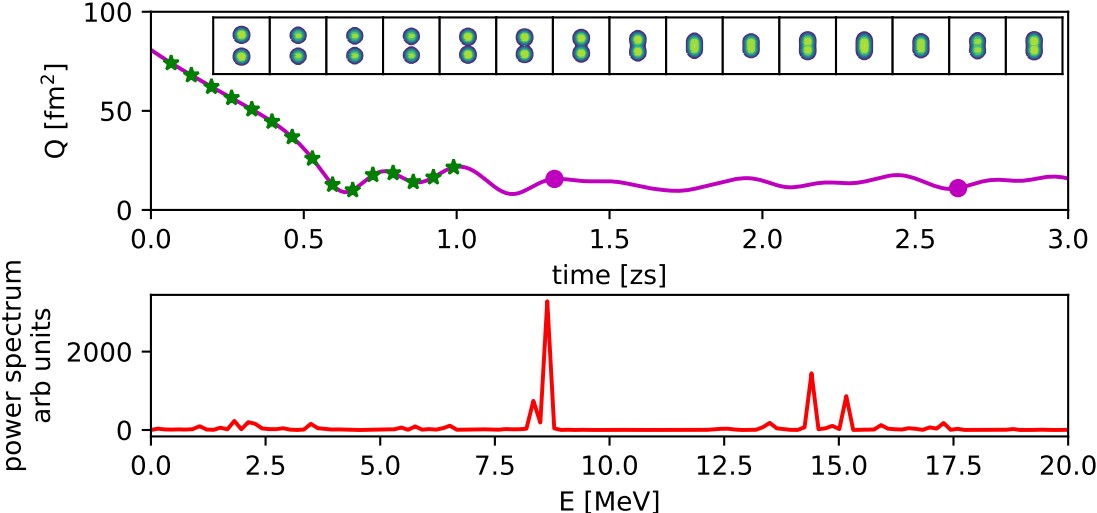

Figure 1: Upper panel shows the quadrupole moment of matter distribution of two colliding alpha particles. Two solid circles show the snapshots after 2000 and 4000 iterations when the wave functions of the $^8$Be$^*$ are taken to be used as starting points in the triple alpha calculations. Starred points correspond to the density snapshots in the inset frames. The lower panel shows the power spectrum of the oscillations of the compound $^8$Be nucleus.

A ground state alpha particle is calculated in static Hartree-Fock to a well-converged solution in a 16×16×16 fm coordinate space box with 1 fm grid spacing in each Cartesian direction. Time-dependent calculations for $^4$He+$^4$He collisions are performed by placing two identical $^4$He ground states separated by a given amount and initialised with instantaneous boost vectors at $t = 0$ which are calculated from a user-defined impact parameter and centre of mass energy of the collision, accounting for the Coulomb trajectory of the reacting $^4$He nuclei as they come from infinity.

At later times during the $^4$He+$^4$He collision, the wave functions of the combined $^8$Be$^*$ nucleus are saved to be used as a starting point for a further TDHF calculation in a larger box. For each specific calculation, the particular parameters used are given as the results are presented in the next section.

## 3 Results

The ground state alpha particle, as obtained form the static Hartree-Fock calculation with the SLy4d interaction has a binding energy of 17.67 MeV, which compares with an experimental value of 28.30 MeV [13]. This under-binding by 40% could clearly have a strong influence on the results, but for the present study a known Skyrme force from the literature was chosen as a baseline for investigation. A separate study with a Skyrme interaction which fits $^4$He better is certainly warranted but not pursued further here.

### 3.1 2-$\alpha$ reactions

To initiate a two alpha particle collision, two $^4$He ground states were placed in a coordinate grid box with dimension 20×16×20 fm with centres at $(0, 0, -4)$ fm and $(0, 0, 4)$ fm. The alphas were given initial boosts to send them travelling towards each other with impact parameter

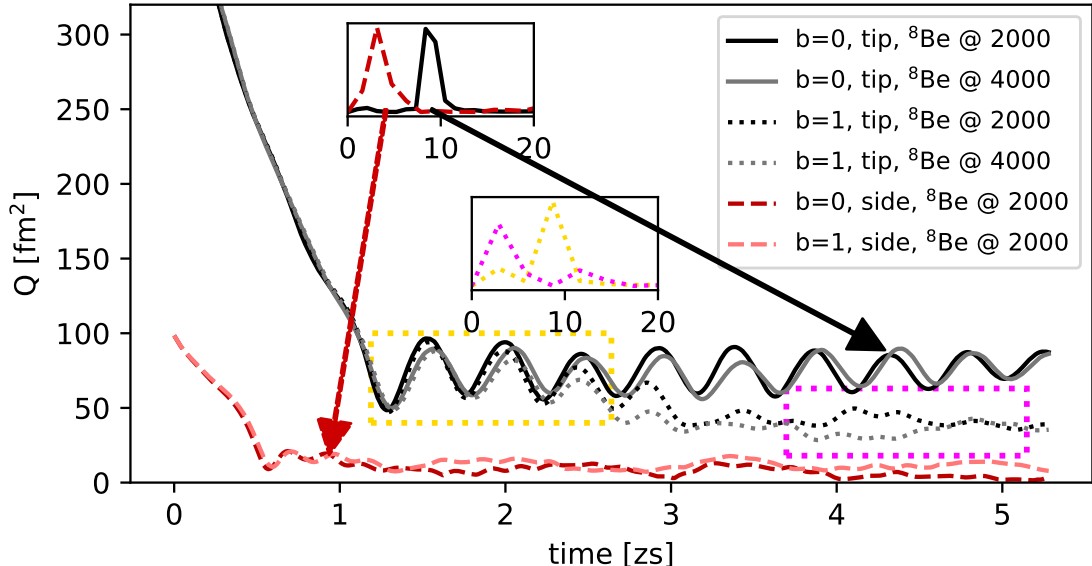

Figure 2: Quadrupole moment of the matter density during collisions of $^4$He with compound $^8$Be$^*$ nucleus with initial conditions as per the legend and discussed in the text. The two insets show the Fourier power spectrum of parts of the quadrupole moment: The upper inset panel shows the spectrum of the resonance created in a tip collision (solid black) and a side collision (dashed red) both at b=0 and using the 2000 configuration of $^8$Be$^*$. The lower panel shows two spectra from the b=1 tip 2000 configuration with the yellow (lighter) dotted line from the time signal as shown in the yellow (lighter) dotted box, and the pink (darker) dotted line showing the spectrum from the later time signal in the pink (darker) dotted box.

$b = 0$ fm and centre of mass energy $E_{CM} = 1.0$ MeV. The nuclei fuse and remain fused for the duration of the TDHF calculation. The quadrupole moment of the matter distribution, defined as

$$Q = \sqrt{\frac{5}{16\pi}} \int d^3 r \rho(r)(2z^2 - x^2 - y^2) \,, \tag{1}$$

where $\rho$ is the total nucleon density in the entire multinucleus system, is shown in Figure 1. Also shown is the Fourier power spectrum of the time signal of the quadrupole oscillations. The time-series for the transformation is sampled starting at 800 fm/c = 2.64 zs, for 2048 data points, which is up to $\simeq 29.7$ zs. The positions of the peaks are rather insensitive to the sampling window. Two states are apparent in the spectrum: One at 8.5 MeV and the other at 14.7 MeV. Both shows some fragmentation, presumably due to artificial discretisation in the coordinate space box [14].

The lowest known $2^+$resonance state in $^8$Be is at 3.03 MeV $\pm$ 10 keV [15]. This is either not probed by the reaction mechanism in the TDHF calculation, or the energy is considerably overestimated. The second experimentally-observed excited state in $^8$Be is a broad $4^+$ resonance at 11.35 MeV [15].

## 3.2 3-$\alpha$ reactions

The parameter space for triple alpha ($^8$Be$^*$+$^4$He) reactions is much larger than for $^4$He+$^4$He: The $^8$Be$^*$ is not spherical, so there will be dependence on the initial orientation of the reacting nuclei. The $^8$Be$^*$ nucleus is not in a stationary state, so there may be dependence upon the

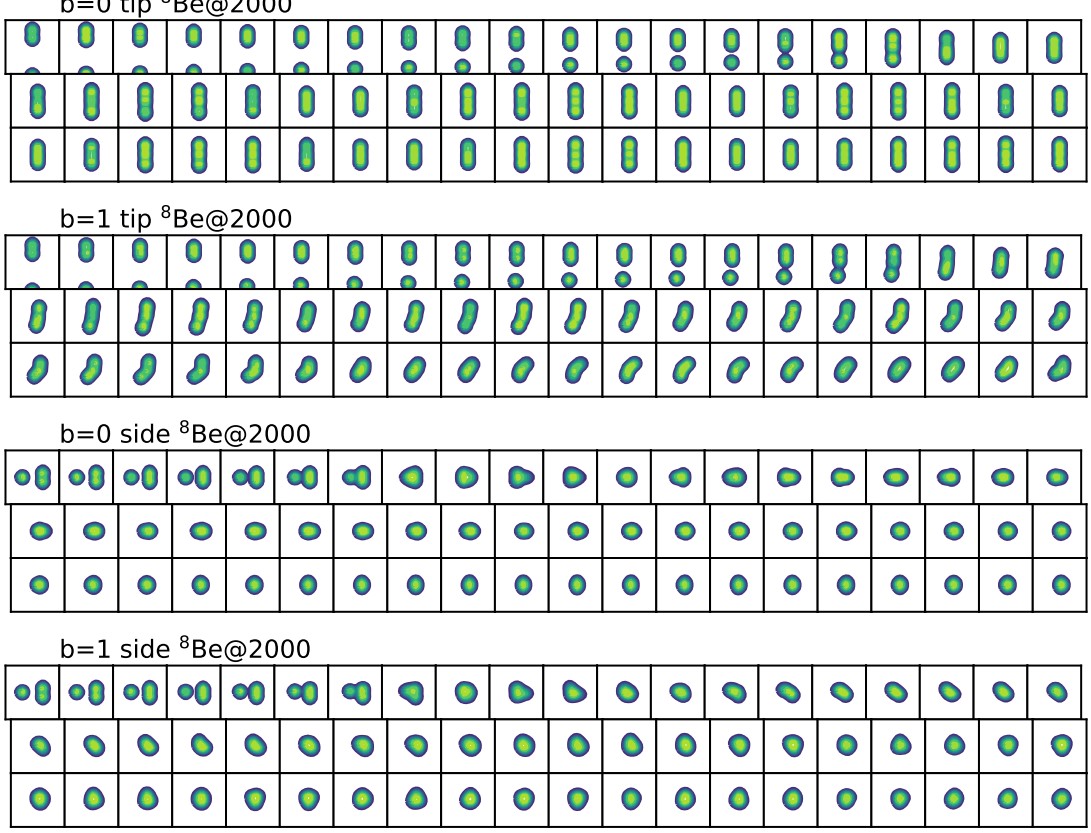

Figure 3: Snapshots of the time evolution of the total density for four of the trajectories shown in Figure 2, as labelled. Snapshots are every 20 fm/c ($\simeq$ 0.067 zs) starting from the top left and reading first left-to right, then top to bottom. The final frames, on the bottom right of each set, are at a time 1220 fm/c ($\simeq$ 4.0 zs).

exact configuration of the $^8$Be$^*$ at the moment of impact. Here, we make a study of these extra parameters, but concede that a much fuller study is needed for a complete picture.

Two different starting configurations are used for the $^8$Be$^*$ nucleus, as indicated by the solid circles on the line in Figure 1. These are somewhat arbitrarily chosen and labelled configurations 2000 and 4000 (because of the number of iterations in the 2-$\alpha$ TDHF calculation), though we note that one of the configurations is near a maximum in the value of $Q$ while the other is near a minimum. Starting orientations are limited to the two extremes of impinging along the long or short axes of the $^8$Be, labelled "tip" and "side" collisions respectively, and with impact parameters selected between $b = 0$ fm and $b = 1$ fm only. The centre of mass collision energy is fixed at $E_{cm} = 2.0$ MeV. This energy is chosen as it is close to the Coulomb barrier, which lies between 1 and 2 MeV, and we are interested in fusion reactions below the threshold for other mechanisms (e.g. fusion-fission). This energy ($E_{cm} = 2.0$ MeV) was also the choice made in a previous TDHF study of the triple-alpha reaction [4], though the results at other energies deserve future study to check the energy-dependence of the reaction mechanism.

Figure 2 shows a summary of the results for simulations up to 5 zs. Figure 3 shows some details of the evolution of the density leading to the results of Figure 2. The b=0 tip configurations lead to a rather stable large-amplitude oscillation which remain in a chain state. Side configurations lead to more compact states with smaller-amplitude oscillations in which triangular configurations appear.

Mixing of the mean-field TDHF configurations via a Fourier spectrum analysis gives an

estimate for the energies of the excited states of $^{12}$C involved. The upper inset panel in Figure 2 shows the power spectrum from the chain state oscillations at around 9 MeV, and from the triangular oscillations at around 4 MeV. The lower inset panel shows that in a b=1 tip collision, the nucleus initially oscillates in the 9 MeV chain state before quickly (∼2 zs) decaying to the 4 MeV triangular state. This interpretation is seen in the snapshots of the time-dependent density in Figure 3, and qualitatively agrees with a previous study [4].

A more sophisticated treatment then mixing via Fourier analysis would be needed to obtain definite spins for each state. Angular momentum projection, followed by the use of time as a generator coordinate to give a basis for mixing Slater Determinants for structure information [16], is our longer term goal to achieve this.

## 4 Conclusion

Time-dependent Hartree-Fock has been used to instigate the triple-alpha reaction, with the dynamics analysed in terms of the energies of vibrational states within the compound nucleus. Identifiable chain and triangular vibrational states at around 9 and 4 MeV respectively are found, with decay from the chain to triangular states occurring with a time dependence on the initial condition.

Perspectives for future study include a mixing of Slater Determinants using time as a generator coordinate, to analyse the spectrum more rigorously, and with the full degrees of freedom that that TDHF calculations afford, rather than measuring only the quadrupole response. A fuller mapping of the parameter space ($E_{CM}$, $b$, orientation), and the form of the nuclear interaction, may afford further insights.

## Acknowledgements

The authors would like to thank the University of Surrey for access to its High Performance Computing facility.

**Funding information** This work was performed with funding from the UK Science and Technology Facilities Council (STFC) under grants ST/P005314/1 and ST/N002636/1. Calculations were performed using DiRAC Data Intensive service at Leicester (funded by the UK BEIS via STFC capital grants ST/K000373/1 and ST/R002363/1 and STFC DiRAC Operations grant ST/R001014/1).

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
