# Peer review of "A time-dependent Hartree-Fock study of triple-alpha dynamics"

_SciPost Physics Proceedings, doi:SciPost Phys. Proc. 3, 047 (2020)_

## Round 1 · Referee Report · Anonymous (Referee 1) · 2019-9-9

Strengths

The paper address the delicate issue of forming carbon 12 from three alphas collisions. They used quite advances microscopic theories.

Weaknesses

This is an intermediate step study and maybe some more systematic study varying input parameters would be desirable

Report

In this article, the authors reanalyze the possibility to obtain a 12C from the reactions of three alphas using the TDHF approach. As mentioned by the authors, such studies were already made in the past, the novelty here being the consideration of a two-step process and the method to extract the excited state energy. Some of the result are interesting and I understand that the present article is an intermediate step towards more systematic studies. I have a few comments that the authors might consider: 1) It is not clear in general if the HF/TDHF approach can describe light systems such as alpha particles or carbon. Can the authors give the ground state energy of these two nuclei obtained at the HF level and also compare with experiments? This would give an indirect hint on the precision one could achieve on energies. 2) Regarding the sentence “The centre of mass collision energy is fixed at Ecm = 2.0 MeV.”. Can the authors motivate physically their choice of energy? If the authors have made tests with different energy, a comment on how the peaks obtained from the spectral analysis depends on the beam energy might be of interest. 3) Another delicate issue is that one usually discuss the formation of the 12C to the Hoyle state that has a definite spin. TDHF simulation leads to a final state which is a combination of different spins. Isolating a given spin can be made with some effort by projecting on total spins at the intermediate and/or final time. Another way would be to use the approximate relation between initial impact parameter and final internal spin. I am not asking the authors to solve this issue, but maybe a comment on this might enrich the article. 4) As a final remark, there are numerous misprints in the text that should be corrected in the final version. I collected some of them in the text: analsis, tracjectory, multinucles, resononance, calcualtion, lablled… As I say above, the calculations are of interest and deserve to be published. I hope the authors will however follow some of my recommendation to improve the manuscript.

---

## Round 2 · Referee Report · Anonymous (Referee 1) · 2019-10-14

Strengths

The article present novel aspects related to the formation of carbon from triple alpha process

Report

The authors have accounted for all the recommendations I asked for in my previous report. For this reason, I recommend the article for publication.

---

## Round 2 · Author Response

We thank the referee for the comments.

Yes, the paper is an intermediate step, though we hope the results as presented are interesting in themselves. They reflect what was presetned at the conference for which the paper is intended as a proceeding article (which also has a 6 page limit, as a poster contribution). I agree with the stated weakness that more variation of the input parameters is needed to give a clearer picture of the validity of this approach

In answer to the particlar points in the report

1) We included a sentence about the calculated (17.67 MeV) vs experimental (28.30) binding energy of He-4 at the beginning of the "Results" section. This serious underbinding is commented on in the same section - that it may be expected to have qualitative effects on the results, but that we wish to start from a known interaction to give a starting point for the study

2) 2 MeV was chosen as being just above the Coulomb barrier which is between 1 and 2 MeV (since 1 MeV is too low to fuse, we found) and we are interested in low-energy behaviour. For much higher energy qualitatively different dynamics would be expected - e.g. fusion-fission. I'd expect the resonances that we saw to be governed more by the properties of the force; with the oscillations' frequencies governed by a restoring force that does not depend too much on entrance channel behaviour except to the extent of how many cycles the oscillation can happen. This is of course speculation until more calculations are performed and analysed, which await future work. In section 3.2 I have expanded the second paragraph which previously stopped at the sentence "The centre of mass collision energy is fixed at E_cm=2.0 MeV" to give a motivation for this energy. I note that the similar previous study by Umar et al. (our [4]) used 2.0 MeV as the collision energy, by coincidence (our value was not chosen because they used it, but for the motivation mentioned above)

3) Agreed. Ultimately we see this work as a stepping stone to something like a time-dependent generator coordinate method (with time as the generator coordinate) from which a spectrum with good angular momentum could be generated from projected and mixed Slater Determinants. The referee's suggestion of using the approximate relation between impact parameter and spin would be a computationally cheap way of getting an idea of the spin, and we'll look at this suggestion for the followup work. An extra paragraph has been added to section 3.2 at the end of the section to discuss obtaining spins.

4) Sorry. Numerous typos changed, and run through a spell checker.

Other changes: * changed the postcode in my affiliation which I had typed incorrectly

  • in the conlcusions I have been more explicit about which parameters deserve further study, in line with the referee's comments.

---

## Round 2 · List of Changes

changes all detailed in the "author comments" field

---

## Editorial Decision

published